# A warm jet in a cold ocean

Jennifer A. MacKinnon [1✉], Harper L. Simmons [2], John Hargrove [3], Jim Thomson [4], Thomas Peacock [5], Matthew H. Alford[1], Benjamin I. Barton [6], Samuel Boury [7], Samuel D. Brenner [4], Nicole Couto [1], Seth L. Danielson [2], Elizabeth C. Fine[8], Hans C. Graber[3], John Guthrie [4], Joanne E. Hopkins [6], Steven R. Jayne [8], Chanhyung Jeon [5,9], Thilo Klenz [2], Craig M. Lee[4], Yueng-Djern Lenn [10], Andrew J. Lucas [1], Björn Lund [3], Claire Mahaffey[11], Louisa Norman[11], Luc Rainville [4], Madison M. Smith [4], Leif N. Thomas[12], Sinhué Torres-Valdés [13] & Kevin R. Wood[14,15]

Unprecedented quantities of heat are entering the Pacific sector of the Arctic Ocean through Bering Strait, particularly during summer months. Though some heat is lost to the atmosphere during autumn cooling, a significant fraction of the incoming warm, salty water subducts (dives beneath) below a cooler fresher layer of near-surface water, subsequently extending hundreds of kilometers into the Beaufort Gyre. Upward turbulent mixing of these sub-surface pockets of heat is likely accelerating sea ice melt in the region. This Pacific-origin water brings both heat and unique biogeochemical properties, contributing to a changing Arctic ecosystem. However, our ability to understand or forecast the role of this incoming water mass has been hampered by lack of understanding of the physical processes controlling subduction and evolution of this this warm water. Crucially, the processes seen here occur at small horizontal scales not resolved by regional forecast models or climate simulations; new parameterizations must be developed that accurately represent the physics. Here we present novel high resolution observations showing the detailed process of subduction and initial evolution of warm Pacific-origin water in the southern Beaufort Gyre.

[1] Scripps Institution of Oceanography, University of California San Diego, San Diego, CA, USA. [2] University of Alaska Fairbanks, Fairbanks, Alaska, USA. [3] Center for Southeastern Tropical Advanced Remote Sensing, University of Miami, Miami, FL, USA. [4] Applied Physics Laboratory, University of Washington, Seattle, WA, USA. [5] Department of Mechanical Engineering, Massachusetts Institute of Technology, Cambridge, MA, USA. [6] National Oceanography Centre, Liverpool, UK. [7] Courant Institute of Mathematical Sciences, New York University, New York, NY, USA. [8] Woods Hole Oceanographic Institution, Woods Hole, MA, USA. [9] Department of Oceanography, Pusan National University, Busan, South Korea. [10] School of Ocean Sciences, Bangor University, Bangor, Wales, UK. [11] School of Environmental Sciences, University of Liverpool, Liverpool, UK. [12] Department of Earth System Science, Stanford University, Stanford, CA, USA. [13] Alfred Wegener Institute, Bremerhaven, Germany. [14] Joint Institute for the Study of the Atmosphere and Ocean, University of Washington, Seattle, WA, USA. [15] NOAA Pacific Marine Environmental Laboratory, Seattle, WA, USA. ✉email: jmackinnon@ucsd.edu

Arctic summer sea ice has been rapidly declining[1–4], with significant implications for global climate through albedo and other feedbacks[5]. In the Western Arctic, sea ice reductions are largest in areas near and just offshore of the Chukchi shelf, where warmer Pacific water enters through Bering Strait[6]. Several studies point to the rapidly rising heat content and throughput of this in-flowing Pacific origin water as a primary cause for accelerating sea ice melt in this region[7,8]. Accurate prediction of sea ice melt rates hence requires a better understanding of the processes and pathways by which warm incoming Pacific water enters the main basin.

Unlike most of the world's oceans, density in the upper Arctic Ocean is primarily controlled by salinity[9,10]. The Beaufort Gyre (BG) is home to the freshest surface waters, resulting from a combination of river inflow, precipitation, and sea ice melt which are consolidated in the center of the gyre through the convergence of wind-drive flows[11]. Strong salinity stratification allows heat to be sequestered sub-surface, in water masses that are warmer but saltier than cool, fresh, surface water.

In the Western Arctic, the largest reservoir of heat in the upper ocean (above ~200 m depth) is Pacific Summer Water (PSW)[12,13]. PSW originates as Pacific ocean water entering through Bering Strait, and is considerably warmed while transiting across the shallow Chukchi shelf during summer months[14]. An appreciable amount of the summer warmed water subducts, allowing heat to spread into the basin interior, where it is sequestered from direct interaction with the atmosphere. Once subsurface, PSW is found throughout the Beaufort Gyre[15]. Here we define PSW to be water with potential density between 23.2 and 25.2 kg m$^{-3}$, warmer than 0 °C, roughly consistent with other recent characterizations[8,13,16]. Within the central BG, PSW is typically found 30–100 m below the surface, frequently in the form of patchy eddies and filaments[13,17,18].

The growing heat content of the PSW sub-surface layer has a first-order effect on accelerating sea ice melt rates in this region, in several ways[6,15,19]. In the short term, the fate of subducted heat may impact the timing of sea ice growth in late autumn, if it remains close enough to the surface to be rapidly mixed upwards by strong fall storms[20]. Longer-term and basin-wide, the observed rate of Arctic sea ice decline thermodynamically requires only a 1 W m$^{-2}$ imbalance in heat exchange between ocean and atmosphere[7]. Annually averaged, heat flux rates through Barrow Canyon were 3 TW in 2010[21], a value that has likely grown in the last decade[8]. If all the heat entering through Barrow Canyon subducted and spread out evenly sub-surface within the BG, that would lead to a warming rate for that layer of 3 W m$^{-2}$[15]. Observations show that the heat content of the sub-surface PSW within the BG has nearly doubled over the last 30 years[15]. If all this heat were turbulently mixed upwards, it could melt more than a meter of sea ice[15].

Pacific origin water that enters the Arctic through Bering Strait brings not only heat, but supplies nutrient-rich water to the Chukchi and East Siberian shelves[22,23]. High productivity combined with enhanced rates of sedimentary denitrification across the shelf[24,25] results in primary productivity in the western Arctic Ocean being typically nitrate limited[26]. The physical processes that control the subduction, stirring and mixing rates of Pacific origin water hence also regulate the supply of nutrients that sustain primary production in the BG. An improved understanding of the PSW subduction process is critical for understanding ecosystem functioning and evolution, especially under changing Arctic conditions.

The primary motivation for this work is thus to observe and identify the processes by which PSW subducts. It may seem intuitive that this incoming warm but salty (denser) water would slide under the fresh (lighter) surface water in the BG. However, water of different densities may stably sit side by side in the ocean when the pressure force associated with density gradients is balanced by the Coriolis force acting on currents that flow perpendicular to density gradients, or along fronts[27]. This type of geostrophic balance is common especially in large-scale oceanic and atmospheric systems. In order for PSW to subduct, something has to break that geostrophic balance. Some PSW may subduct through wind-driven Ekman pumping along the edge of the Chukchi shelf, analogous to ventilation of the thermocline at mid-latitudes[13], though more recent work argues this effect is modest[28]. However the warmest water that subducts to enter the BG travels through Barrow Canyon in late summer as a surface-intensified current[16,29,30]. Some of the Barrow Canyon outflow has been observed flowing in currents that hug the continental slope towards both the East and West, depending on complex combinations of wind and buoyancy forcing[31–33]. Those currents may in turn be subject to instabilities that facilitate exchanges with the basin interior[33–35]. Yet the process of warm PSW subduction in the Arctic has not been previously observed in detail. Here we present observations of instabilities and warm PSW subduction occurring directly downstream of the Barrow Canyon outflow, and argue that the process of subduction is directly related to meanders in that outflow.

Accurate prediction of the effect of PSW on Arctic sea ice melt and evolving ecosystems requires a better understanding of the physical processes that mediate subduction, stirring and mixing of that water once it enters the Arctic Ocean. The observations presented here show a subduction process with lateral scales of km or smaller. These so-called submesoscale processes cannot be well represented even with high resolution eddy-resolving models (grid scales of 2–3 km, e.g.,[36]), let alone the much coarser regional or global forecast models[37]. Though there have been some initial attempts to parameterize the subduction process[38], further development is limited by a lack of clear understanding and observational verification of the physical mechanisms involved. The U.S. Office of Naval Research Stratified Ocean Dynamics of the Arctic (SODA) project was designed to explore and understand the evolving upper ocean stratification of the Western Arctic ocean. The novel measurements reported here from SODA scientists and collaborators represent the most detailed observations to date of the subduction process.

## Results and discussion

**Observations of a meandering warm jet**. Satellite observations from 15 Sept 2018 show a jet of warm water meandering offshore from Barrow Canyon (Fig. 1b). The warmest surface waters in this image are observed directly over Barrow Canyon. The jet initially turns eastward, but then makes a sharp turn northward/offshore near an indentation in the topographic slope (blue contour). Subsequent meanders in the jet of ~50–100 km scale are consistent with previous observations[39,40]. The Sea Surface Temperature (SST) anomaly associated with the jet is visible with reduced amplitude over 100 km offshore, impinging upon a remnant patch of multi-year sea ice. Convoluted stirring patterns can be seen between the warmest waters of the jet and cooler Arctic surface water. Upper ocean currents measured during the ship survey (Fig. 1c, arrows) are consistent with the meandering pattern visually apparent in SST, with peak amplitudes of over 1 m s$^{-1}$ in the core of the jet.

Uniquely high-resolution ship-based profiling (see Methods) reveals the complex sub-surface structure of the meandering jet (Fig. 1c). At the onshore edge of the observed jet, water warmer than 4 °C extends from the surface to more than 100 m depth. On the northern/offshore edge (far right of the lowest portion of Fig. 1), the vertical extent of warm water compresses to less than

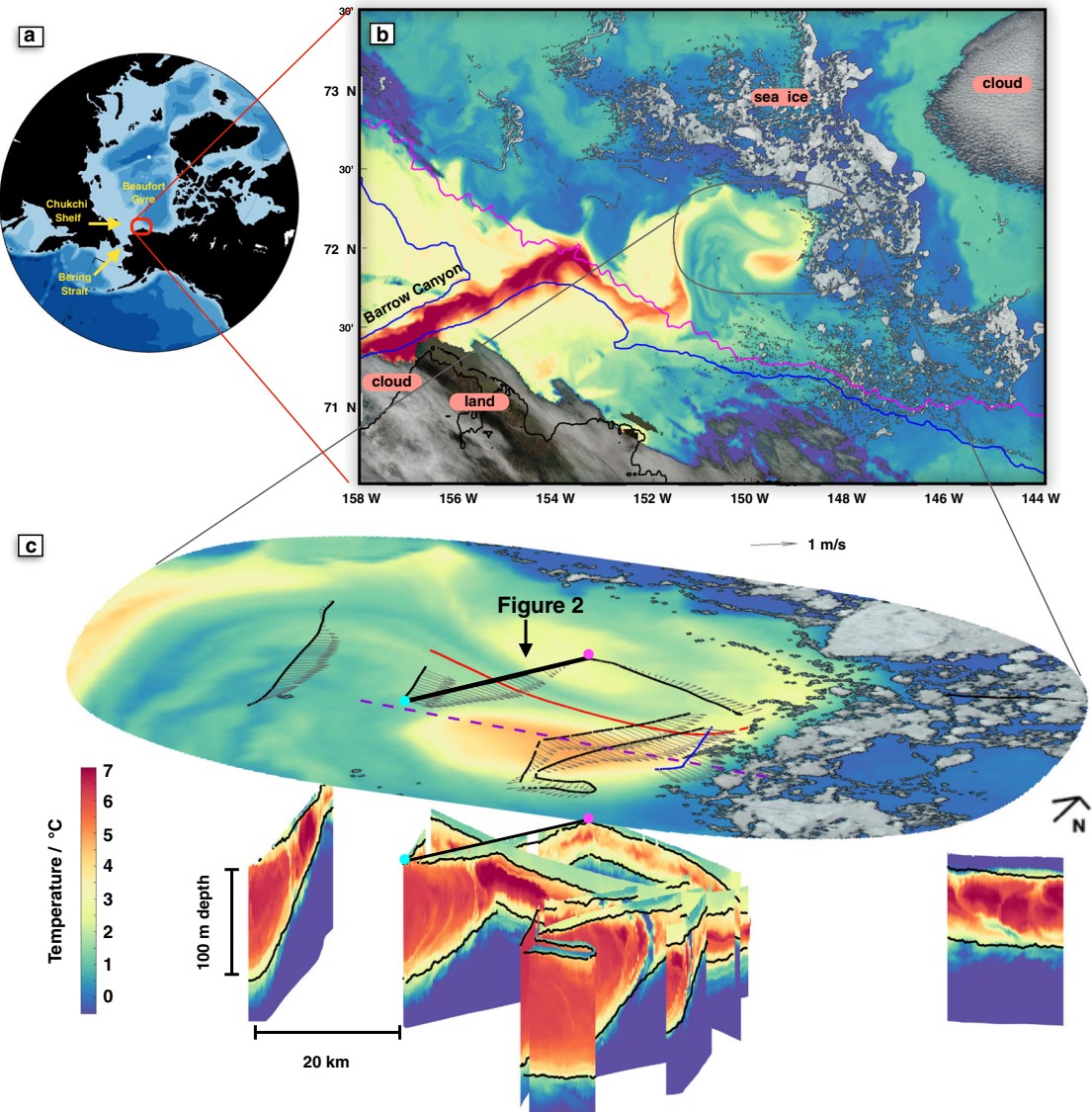

**Fig. 1 Overview of warm jet observations. a** Red square shows the location of the subsequent panel in the Pan-Arctic context. **b** Hybrid MODIS satellite image from 15 Sept 2018 showing sea ice and clouds in true color, and sea surface temperature (SST) over open water. The black line is the Alaskan coast, and colored lines are the 100 m (blue) and 1000 m (magenta) isobaths. **c** Expanded view of part of the satellite image (above) and sub-surface temperature measurements taken 14–17 Sept 2018 from multiple instruments (below). Pathways for FastCTD (black line), Wirewalker (red), and Microstructure profiling (blue) are shown on the surface image, as are observed ocean currents (vectors) averaged over the top 90 m, scale arrow to the right. The pathway of the second FastCTD survey 9 days later is shown with a dashed purple line. For the sub-surface temperature data, the 23.2 and 25.2 kg m$^{-3}$ potential density surfaces are contoured in black. Surface images and sub-surface measurements share a common temperature scale at lower left. The location of the section shown in detail in Fig. 2 is indicated in **c** here with a thicker black line and magenta/cyan dots on the northern/ southern ends.

30 m, centered around 40 m below the surface. The sloping wedge of PSW is overlain by a thin layer of cooler, fresher surface water. On the north-eastern edge of the survey, within the marginal ice zone, the PSW layer is completely sub-surface and appears to have partially broken up into a series of closely spaced sub-surface warm patches.

A cross-sectional slice reveals the internal structure of the warm-water jet in greater detail (Fig. 2). The core of the jet is bounded by the two isopycnals used for our definition of PSW (Fig. 2a, bold white contours). Between the highlighted density contours and in the main jet (cross-jet distance of less than 17 km), the water has an average temperature of 6 °C, and a salinity of 30.9 psu. Velocity in the along-jet direction peaks sub-surface and is concentrated between the highlighted density surfaces (Fig. 2c).

The context for this unusually warm water can be seen by comparing its temperature and salinity values to other regional water masses (Fig. 3). As temperature and salinity are both largely conserved along water pathways in the ocean interior, Temperature–Salinity (T–S) plots are often used to classify water masses of different origins. They are also used to identify evidence of mixing between water masses, which manifests on a T-S plot as data on a straight line between end-point water masses[41]. The water flowing through Barrow Canyon in late summer (Fig. 3a) shows both typical surface intensified warm PSW and cooler slightly saltier Pacific Winter Water below[19]. The warm water from Fig. 2 can be identified in T–S space as the incoming PSW, which is mixing with both winter water below and cool-fresh melt-water above (Fig. 3b).

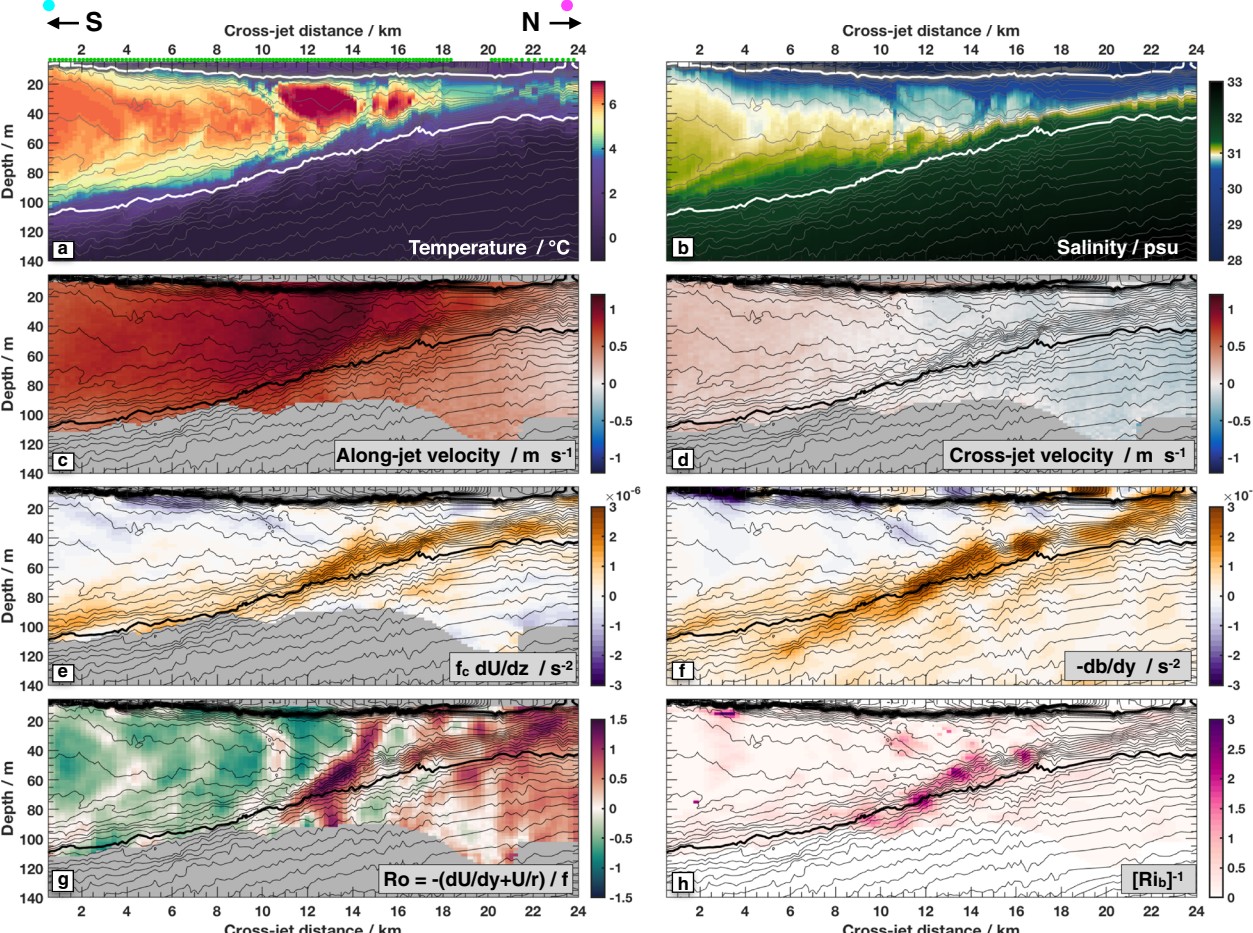

**Fig. 2 A slice through the meandering jet taken on 15 Sept 2018 (location shown in Fig. 1c ).** On all panels, isopycnals are contoured every 0.1 kg m$^3$ (thin black lines), with the 23.2 and 25.2 isopycnals in bold white or black. Color shading in each panel represents **a** temperature; **b** salinity; **c** along-jet velocity, which has been rotated 23.4° southward of East; **d** cross-jet velocity; **e** vertical derivative of along-jet velocity scaled by the inertial frequency adjusted by the jet curvature ($f_c$, Methods); **f** cross-jet buoyancy gradient; **g** the Rossby number, Ro, a non-dimensional measure of the relative vorticity (Methods), **h** inverse Richardson number, Ri$_b$, calculated from the lateral buoyancy gradient assuming thermal wind balance (Methods). Both Ro and Ri$_b$ take on values with magnitudes close to one indicating that the flow follows submesoscale dynamics. The green dots above **a** show the location of each FCTD profile, which are interpolated between for plotting purposes. Arrows above **a** indicate rough Northward and Southward directions, and the magenta/cyan dots are also shown in Fig. 1c, for orientation.

To first order this meandering jet appears to be in geostrophic balance; lateral pressure gradients, set by a combination of sea surface slope and lateral density gradients within the water column, show similar patterns to those of the Coriolis force acting on the along-jet flow. Since we do not have a direct measurement of pressure associated with sea-surface height gradients, the balance is assessed by comparing the vertical gradients of both forces, known as thermal wind balance (Methods, Fig. 2e, f). In this case, the curvature of the meandering warm jet visible in Fig. 1 adds a significant term to the effective Coriolis force (Supplementary Methods).

The strength of these currents puts this feature firmly in the nonlinear realm. Much of the instability theory considered by previous work has been based upon small perturbations to modest amplitude currents. The non-linearity of this current system may be quantified by the strength of both the lateral and vertical shear. Lateral shear and curvature in the along-jet velocity create negative (clockwise) relative vorticity on the southern side of the jet, with positive (counterclockwise) relative vorticity on the northern side (Fig. 2g). This relative vorticity is of roughly the same side as the earth's rotation rate; their ratio gives a non-dimensional Rossby number, which is often used to measure non-

linearity. The component of vertical shear associated with this jet is also strong; normalizing by stratification gives a non-dimensional Richardson number of order one (Fig. 2h). The strength of this laterally and vertically sheared current puts it firmly into the category of submesoscale ocean dynamics[42,43]. Order one values of both Rossby and Richardson numbers are associated with increased propensity to various instabilities that may enhance stirring and mixing of PSW, as discussed further below.

**Subduction processes**. The sub-surface temperature structure visible in Figs. 1c and 2a showcases several steps of the subduction process. Here we loosely interpret the left to right (South to North) variability in these sections as time evolution connecting the more depth-uniform PSW flow that emerges from Barrow Canyon with the discrete sub-surface PSW features visible in the far lower right of Fig. 1c. First, the warm PSW wedge appears to be squeezing sideways (northward in Fig. 2a) and vertically compressing. Second, it is developing a particular pattern of relative vorticity. Within the main wedge of PSW (southward of ~16 km cross-front distance in Fig. 2g), there are alternating

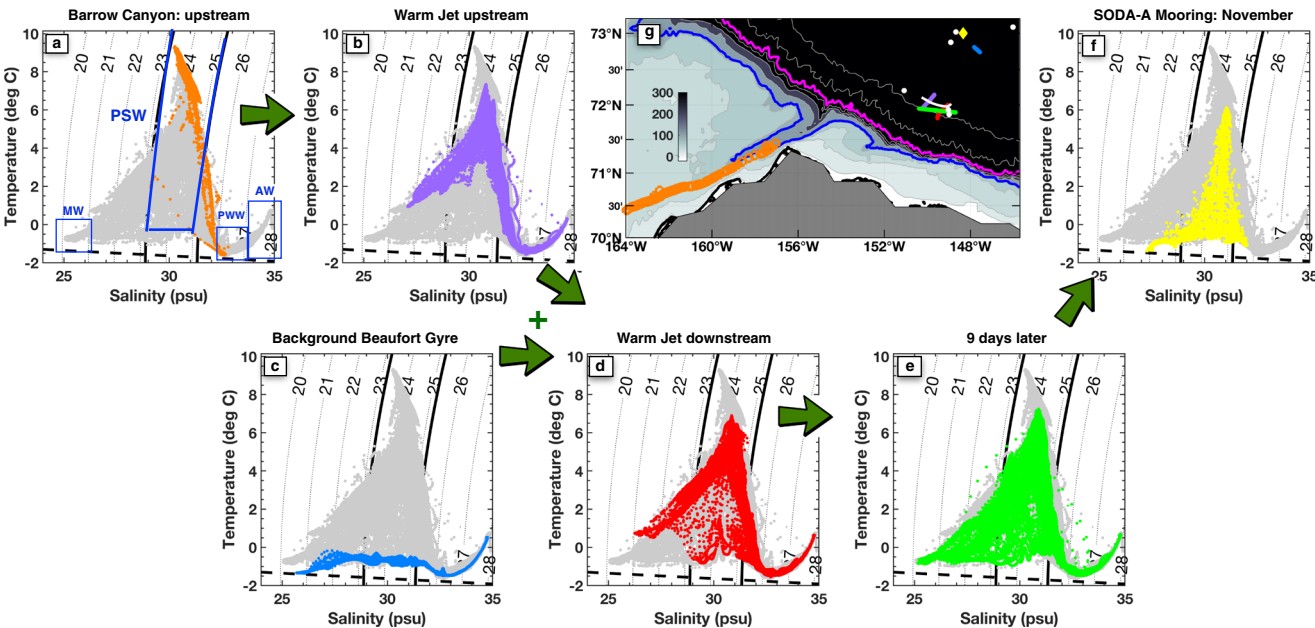

**Fig. 3 Context and evolution of Pacific Summer Water (PSW). a** Temperature and salinity characteristics of water transiting through Barrow Canyon from profiling floats and gliders (orange dots, see "Methods"), the location of which are shown in **g**, and all observations (gray dots, same in all panels). Isopycnals are contoured every 1 kg m$^{-3}$ and the 23.2 and 25.2 kg m$^{-3}$ isopycnals that bound PSW are in bold. Labeled water masses in **a** include Pacific Summer Water (PSW), cool fresh Melt Water (MW), Pacific Winter Water (PWW), and Atlantic Water (AW), expanded discussion in Supplementary. The freezing point of seawater is indicated with a dashed black line. **b** Fast CTD data from the cross-jet survey section shown in Fig. 2. **c** Fast CTD data of typical background Beaufort Gyre water measured on 18 Sept. **d** Fast CTD data from a slightly downstream section through the meandering warm jet, reflecting stirring and mixing between **b** and **c**. **e** Fast CTD data from a subsequent survey through the same region 9 days later on 24 Sept (Fig. 5). **f** Data from the SODA-A mooring measured from 25 Oct–15 November 2018. **g** Map showing locations of all measurements, colors match those in each panel; the 100 m (blue) and 1000 m (magenta) isobaths are contoured, as in Fig. 1. The white line and dots indicate the locations of the Wirewalker drift and CTD stations respectively, data for which is shown in Fig. 7.

broad regions of negative (green) relative vorticity and narrow semi-vertical stripes of positive (red) relative vorticity. Physically, parts of the PSW wedge are starting to spin at different rates relative to each other. Finally, the wedge of PSW is breaking up into discrete sub-surface eddies, as seen in curved, scalloping anomalies of temperature and salinity that line up with the relative vorticity features.

Though sub-surface PSW eddies and filaments have been observed previously, and discussed in theoretical literature, the subduction process has not been observed in detail. Here we discuss each of these observations in light of relevant dynamical processes and describe how they are linked together. We build upon previous work discussing Arctic eddy formation through baroclinic or frontal instabilities[20,44–49], though again we comment that the strong magnitude of the meandering jet here puts it firmly into a nonlinear, or submesoscale regime, distinct from most previous work in this region.

The first step in the observational evolution of PSW is a squeezing and vertical compression of warm PSW (Fig. 2a). Such behavior is reminiscent of previous theoretical work on meandering jets[50]. For a large-scale meandering current, the cross-jet flow is expected to be confluent (laterally squeezing together) upstream of each trough in the meander[45,51]. The section shown in Fig. 2 was taken just upstream of a trough in the large-scale meandering path of the jet (Fig. 1). In this section, the observed cross-jet velocity (Fig. 2d) is indeed confluent, with northward (pink) velocity on the southern side and southward (blue) velocity on the northern side.

Geometrically, confluent cross-jet flow tends to steepen sloping isopycnals, pushing them away from a state of geostrophic balance. Theoretical work shows that the imbalance leads to a series of adjustments that cumulatively tend to restore the flow to

geostrophic balance, which is a preferred, stable state for this type of dynamical system. In particular, theory predicts that a modest, cross-front circulation develops which restores the original isopycnal slope[52]. The anticipated cross-jet circulation can be calculated using the omega equation (see "Methods"), which represents this multi-step process[53].

The inferred cross-front circulation associated with that confluence is shown with contours and arrows in 4e, f. Unlike previous studies of individual sloping fronts, here there are two signs of isopycnal slopes, making a double-front. On the upper side of the PSW wedge, isopycnals slope downward and northward, while below they slope upward and northward. The double-front set-up leads to two calculated overturning cells, a deeper clockwise cell (dotted cyan in Fig. 4e, f) and an upper counterclockwise cell (solid cyan). These cells conspire to push warm water further northward (positive cross-front direction) into the wedge (cyan arrows on Fig. 4e) and spread cool fresh surface water southward atop the subducting jet (pink arrows near surface). The circulation vertically compresses the wedge of warm water from both above and below—the top is pulled down while the bottom is pushed up. The predicted vertical velocities are of order 10 m per day and predicted cross-jet velocities are a few km per day.

While the observations are qualitatively consistent with the calculated double-cell cross-front circulation, the calculated circulation may be an under-estimate given the strength of these currents. The theory underlying the cross-jet circulation shown in Fig. 4e, f formally requires weak currents. For the observed strong flows (order one Rossby and Richardson numbers, Fig. 2) the calculated cross-jet circulation is probably a lower bound[54]. Furthermore, for strong negative relative vorticity ($Ro < -1$, "Methods") a fast-growing, energetic and time-dependent process

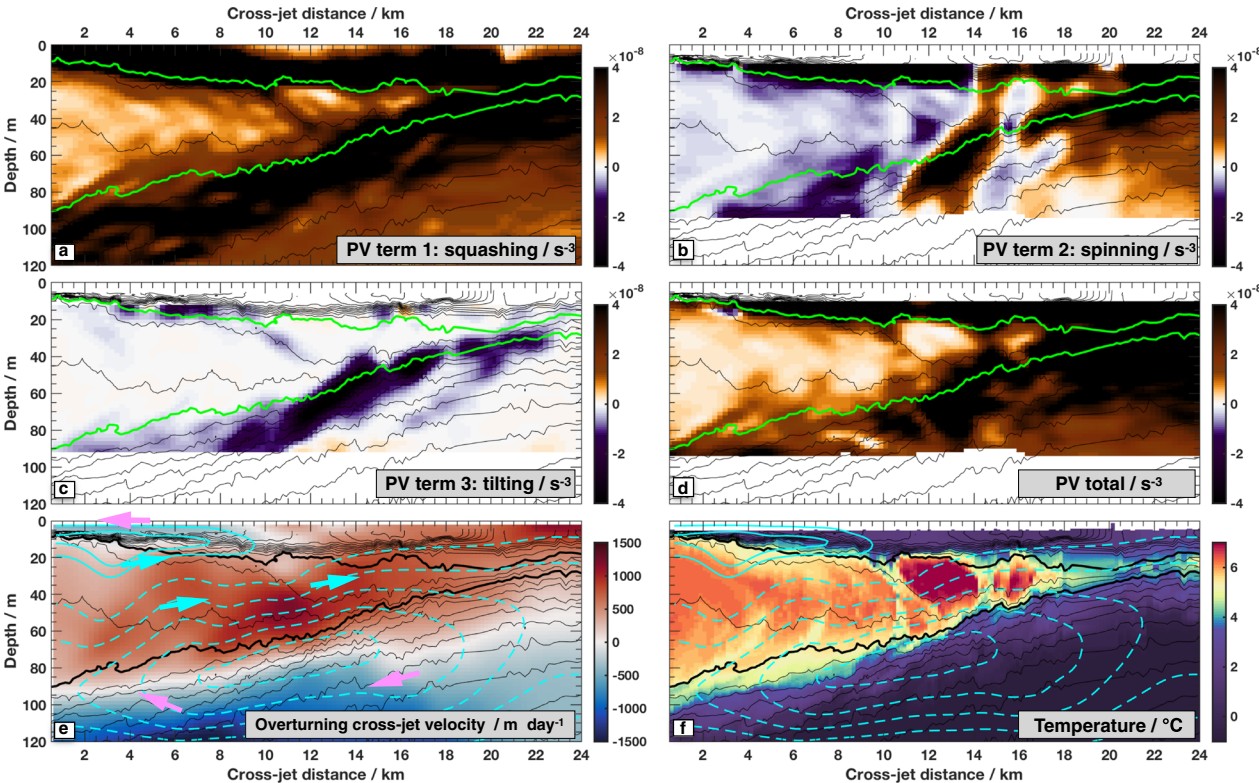

**Fig. 4 Additional calculations for the section shown in Fig. 2.** On all panels, isopycnals are contoured every 0.2 kg m³ (thin black lines), with the 23.2 and 25.2 isopycnals in bold green or black. Color shading in each panel represents **a** the squashing (or stretching) component of potential vorticity (see Methods, where positive values indicate squashing with isopycnals spaced closer together); **b** the relative vorticity (spinning) component of potential vorticity; **c** the tilting component of potential vorticity; **d** the sum of all three potential vorticity components; **e** calculated overturning stream function from the omega equation ($\Psi$, see "Methods"), contours plotted from $-0.5$ to $0\ m^2\ s^{-1}$ with 0.1 increment (dashed) and 0 to $0.05\ m^2\ s^{-1}$ (solid) with 0.02 increment, arrows indicate the direction of each overturning cell; **f** Temperature with the same overturning streamfunction contours as in **e**.

known as centrifugal instability may develop[55,56]. There are a few locations on the section shown in Fig. 2 and on the section just upstream where Ro is close to $-1$ (Fig. 2g). There may be even stronger negative relative vorticity upstream, immediately after the flow exited Barrow canyon (Fig. 1). Evidence for this instability is discussed further in the Supplementary material.

Near the surface, different processes may directly spread cool fresh surface water over the top of PSW. Viewed from above, there is a taffy-like stirring pattern between warmer (coastal origin) and cooler (offshore origin) surface water (Fig. 1). The near-surface water visible in Fig. 2 is one slice through this structure. Near-surface water experiences direct wind and wave forcing and higher levels of turbulence. The combination of that forcing differentially advects fresh near-surface water in a way that is partially decoupled from the stratified ocean below, with a net effect of spreading out freshwater filaments (Supplementary).

The second step in the observed process is the development of the striped patterns of relative vorticity, indicating differential rotation rates, within the PSW wedge (Fig. 2g). These developing patterns are consistent with conservation of a quantity known as Ertel potential vorticity (PV)[42,51] (Methods). Ertel PV consists of three primary components: a vertical "squashing" term (Fig. 4a), a relative vorticity "spinning" term (Fig. 4b) and a "tilting" term (Fig. 4c). Here we include curvature of the meandering jet flow in the relative vorticity term[57–59]. As PSW is squeezed sideways into the wedge of sloping isopycnals, it is vertically compressed, increasing the squashing component of PV (Fig. 2a). This leads to enhancement of the spinning component of PV within the PSW wedge (Fig. 2b), in order to conserve total PV (Fig. 2d). Interplay between the different spatially varying components of this

conserved quantity (Fig. 4) likely produces the observed spatial patterns in relative vorticity. A further step in the water mass evolution may be seen in the furthest offshore observations, on the far lower right of Fig. 1; here PSW is visible in more distinct features, which may be the precursors of individual intrathermocline eddies[18]. The horizontal length scales of the developing relative vorticity patterns, and eventual intrathermocline eddies, are thought to be set by geometric and dynamic constraints as those features adjust back towards a state of geostrophic balance[48,50].

**Initial evolution of sub-surface PSW.** As warm PSW subducts and vertically compresses, the developing patterns of relative vorticity lead to differentiation and discretization of water mass features in Fig. 2. For example, the noses of two scalloping temperature and salinity anomalies can be seen around 50 m depth and 5 and 7 km in the cross-jet distance coordinate in Fig. 2a, b. They extend both downward and southward, and in more muted form upward and southward. These wispy features involve water that is cooler but fresher than the PSW to either side. A more pronounced scalloping feature is centered around 50 m depth and 10.5 km cross-jet distance (Fig. 2a, b), similarly with water that is cooler but fresher than that on either side. In all cases, the temperature and salinity anomalies have compensating effects on density, as evidenced by the small deviations of the contoured isopycnal lines in both panels. Such temperature and salinity anomalies cannot result from vertical motions, as the water below the jet is cooler but saltier. Instead, we posit that the water mass anomalies result from stirring along isopycnals at the edges of this

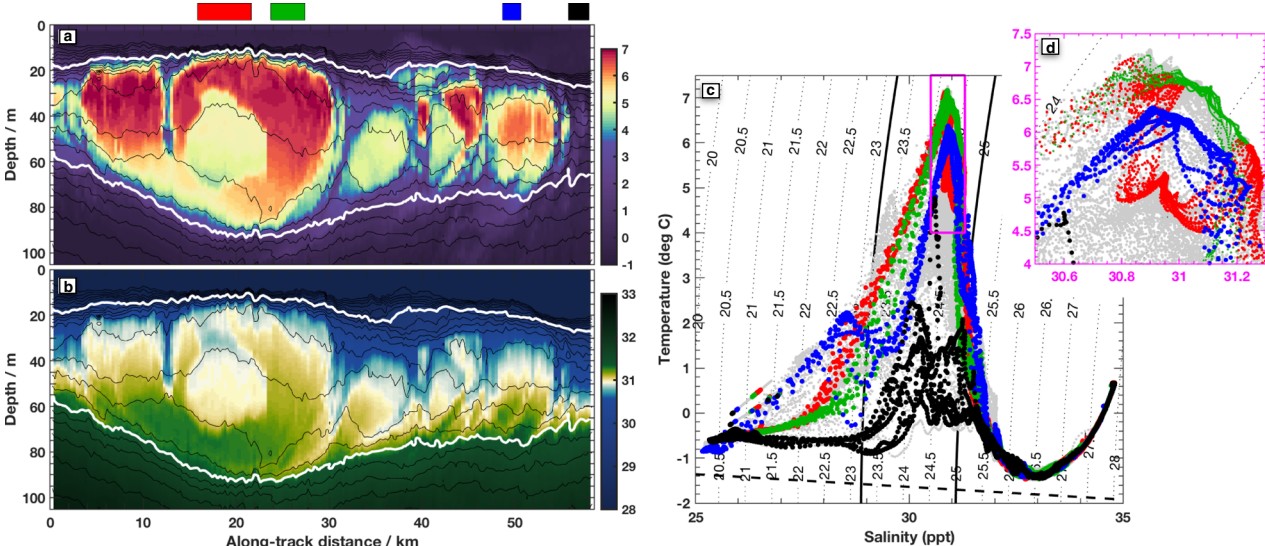

**Fig. 5 Results from a FastCTD survey on 24 Sept, 9 days after the original measurements.** The survey track is shown in Fig. 1c (purple dotted), and took 7 h to complete. **a**, **b** Temperature and salinity sections, as in Fig. 2. The four colored blocks above indicate the subsets regions periods used for subsequent panels. **c** Temperature-salinity measurements as in Fig. 3 for all data in this section (gray) and the four sub-regions indicated with color. **d** Expanded view of the portion of the T-S plot indicated with a magenta box in **c**.

meandering jet, in and out of the page from this section perspective. These water mass anomalies line up with the relative vorticity anomalies (Fig. 2g), providing strong evidence that the developing relative vorticity features described above are responsible for the lateral stirring.

The relationship between lateral stirring and the temperature or salinity anomalies in Fig. 2 is further elucidated by turning back to the T-S diagrams (Fig. 3). When PSW enters the BG (Fig. 3, panel b) in the form of this warm jet, it meanders through water characterized by typical regional temperature and salinity vertical profiles (panel c). Those background profiles show a straight mixing line between surface melt-water above (left side of Fig. 3c), and Pacific winter water below (right side of panel c). The developing relative vorticity features described above stir and mix PSW with the background water along isopycnals, between the peak of the triangle shown in panel b, and the lower leg of that triangle shown in panel c, especially in the isopycnal range bounded by the two solid black lines identifying PSW density. That stirring fills in the T-S triangle, as can be seen in a section further downstream (Fig. 3d). The scalloping features visible in Fig. 2a, b are the initial stage of this stirring process.

During a subsequent survey of the same region 9 days later, the subduction process appears to be complete, with cool fresh water found everywhere near the surface (Fig. 5a, b). Compared to the original survey, mixing and stirring along and perhaps across isopycnals has now filled in the entire triangle in T-S space (Fig. 3e). The stirring process can be seen in more detail by considering four representative profiles through the second survey (Fig. 5 colors). There is some indication that subducted PSW (green profile) is mixing with background water along the same isopycnal found just outside the eddies (e.g., black profile). Similarly, it appears as though the red and blue profiles have exchanged water properties near the temperature maximum (Fig. 5d). The T-S profiles and the bite taken out of the eddy in the red profile section of the snapshot (Fig. 5a) suggest an intriguing form of eddy cannibalism induced by complex three-dimensional velocity and vorticity fields.

**Heat budgets and long term evolution.** Here we consider both the evolution of oceanic heat within this particular event and the

implications for the basin as a whole. Combining observed temperature and velocity in the primary survey gives a lateral heat transport of 35-40 TW for the sections shown in Fig. 1 ("Methods"). This heat flux is probably an underestimate of the true flux, as our measurements do not include the southern edge of the jet or the more complex eddying meanders visible in the satellite image. These heat flux estimates are consistent with but near the high end of previous measurements of peak late summer heat fluxes through Barrow Canyon, consistent with warming PSW inflow[16,21].

For the subduction event described here, some incoming heat is lost to the atmosphere. Over the course of the initial survey (14–17 Sept) average ship-measured heat flux from the ocean to the atmosphere was 122 W m$^{-2}$, similar to previous measurements[60] (Supplementary). As a simple scaling, a heat loss of 100 W m$^{-2}$ cools a 10m surface layer ~0.2 °C/day, which is consistent with what was observed by floats transiting Barrow Canyon over ~10 days. It cannot explain the evolution of the jet in TS space, which must come principally from isopycnal stirring and mixing with background waters of the Beaufort Gyre (Fig. 3). During the early stages of the subduction process, when PSW is near the surface, the measured upward turbulent heat fluxes within the ocean are large, approaching the rate of surface heat loss to the atmosphere. However, once the PSW core has been sequestered further below the surface, measured turbulent heat fluxes are less than 5 W m$^{-2}$ (Supplementary). By the second survey on 24 Sept, air–sea heat fluxes were only 16 W m$^{-2}$ from ocean to atmosphere. The net picture is one of warm water steadily being pulled sub-surface, losing a small portion of its heat upwards though mixing and then air-sea heat loss. As a rough upper bound, integrating a heat flux to the atmosphere of 100 W m$^{-2}$ over the roughly 100 km × 100 km area shown in Fig. 1c gives a net loss of 1 TW, a small portion of the 40+ TW entering the Arctic in this jet. We conclude that the majority of the heat in this event subducts, where it is subsequently sequestered from direct contact with the atmosphere or sea ice. Given that typical maximum winter mixed-layer depths in the BG are only 30-35 m[61], most of this subducted heat may be insulated for months if not longer.

As subducted PSW spreads into the main gyre, the net effect on sea ice will depend on both the lateral stirring rate and the

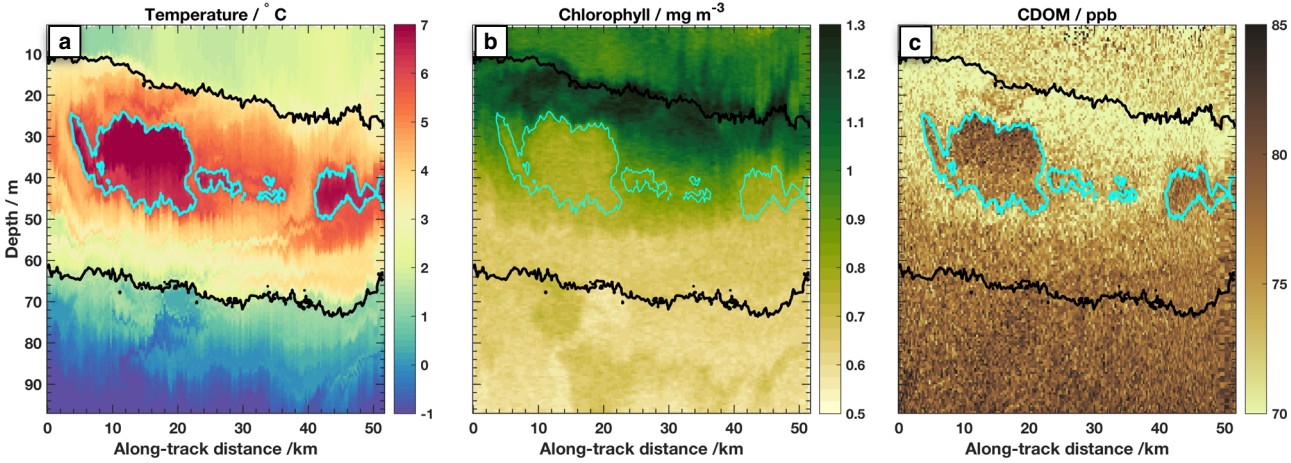

**Fig. 6 Biogeochemical signatures of Pacific Summer Water (PSW): the section view.** Observations along the Wirewalker drift of **a** Temperature, **b** Chlorophyll concentration, and **c** Chromophoric Dissolved Organic Matter (CDOM). CDOM is presented in parts-per-billion (ppb) based on factory calibration of the sensor. In each panel the 23.2 and 25.2 kg m$^{-3}$ isopycnals are contoured in black, and the 6 °C isotherm is contoured in cyan.

upward diffusion rate of heat. Some water is entrained into eastward or westward boundary currents. For example, sub-surface PSW with the same T-S characteristics as observed here was observed flowing westward in the Chukchi Slope Current[62]. Longer term, both PSW subducted directly from the Barrow Canyon outflow, and the portion subsequently caught up in boundary currents, may be drawn into the central Beaufort Gyre and beneath the main ice pack through a combination of advection by the anti-cyclonic circulation and eddy stirring[63]. Water with similar T-S properties was observed passing by the SODA-A mooring, 120 km to the north-east, in early November 2018 (Fig. 3f). Warm water (>3 °C) was observed at the mooring in a series of pulses from 28 October–13 November, between 45 and 90 m; no measurements are available higher in the water column. The maximum temperature observed at SODA-A is slightly reduced compared to the ship survey(Fig. 3f), perhaps by continued stirring and mixing. By November this region was fully ice covered; these measurements thus showcase persistence of very warm water lurking beneath a frozen surface.

**Ecosystem implications.** In addition to carrying sub-surface heat into the basin, the warm PSW carries important biogeochemical tracers that reflect the meandering jet's Pacific origins and transit across the Chukchi shelf in the summer months prior to the field program. Within the warm jet, biogeochemical properties display complex patterns that mirror the intricate subduction, stirring and mixing that controls heat distribution. The profiling, drifting Wirewalker data shows clear signatures in both Chlorophyll concentration and CDOM (Chromophoric Dissolved Organic Matter, Fig. 6b, c) that follow the warmest temperatures (Fig. 6a). CDOM is elevated in a range that maps well onto the 6° C iso-therm (contoured cyan), reflecting both the substantial river influence for the warmest incoming PSW[64], and potentially also scouring as water travels along the Chukchi shelf and through Barrow Canyon.

Surface waters and the upper layer of the plume are nitrate-depleted (<0.1 mmol m$^{-3}$), reflective of algal uptake across the shallow shelf, though some nitrate may be restored through turbulent fluxes within Barrow Canyon[30]. As mentioned above, the stirring of near-surface waters responds to additional wind and wave forcing, so the surface layer may deferentially advect compared to the sub-surface plume. Nitrate concentrations increase within the bottom 20–30 m of the plume and reach ~5

mmol m$^{-3}$ at its base (the 25.2 kg m$^{-3}$ isopycnal) (Fig. 7a). The top of the nitracline is thus embedded within the plume.

Chlorophyll-*a* concentrations are elevated throughout the upper layers of the plume and in the sunlit surface waters, but especially at a depth that maps well onto the top surface of the PSW jet (Figs. 6b, 7b). The depth of the top of the plume and the chlorophyll maximum, deepen in time as the jet flows northward and subducts downward, carrying with it and redistributing shelf-origin waters. Within the upper, nitrate-depleted layers biomass, specifically particulate carbon and particulate organic nitrogen (PC and PON, respectively), is more than twofold higher in the jet (62 ± 29 µg L$^{-1}$ PC and 12 ± 6 µg L$^{-1}$ PON, respectively) compared to the surrounding Beaufort Gyre (26 ± 3 µg L$^{-1}$ PC and 5 ± 0.5 µg L$^{-1}$ PON, Fig. 7d, e and Supplementary). Evidence of enhanced phytoplankton growth within the jet is also found in the stable carbon and nitrogen isotope signatures (Fig. 7f, g and Supplementary). Carbon isotopes of PC are ~2‰ lower in the jet (−26.2 ± 0.8‰) compared to the surrounding Beaufort Gyre (−28.8 ± 0.1‰), indicating enhanced primary productivity[65]. Further, nitrogen isotopes of PON are enriched in the jet (6.7‰) compared to the surrounding Beaufort Gyre (4.6 ± 3.2‰), indicating complete assimilation of isotopically-enriched nitrate within the jet, reflective of the Bering–Strait–Chukchi pathway[24,25] and incomplete nitrate assimilation in the surrounding waters.

Algal growth and biomass accumulation beyond that which has been exported off the shelf within the plume is likely to be severely nitrate limited. However, lateral stirring induced by the jet's developing relative vorticity (Fig. 2g) leading to vertical shear and enhanced turbulent kinetic energy dissipation[18] may drive fluxes of new nitrate across the nitracline and alleviate nitrate limitation within the euphotic zone. Algal growth within subsurface chlorophyll layers makes an important, yet still poorly constrained contribution to Arctic primary production[66,67]. Thus the capacity of this shallow warm meandering jet to carry carbon-laden waters from the shelf into the carbon-poor basin[68] whilst also providing a mechanism for vertical nutrient fluxes may play an important role in biological production across the Canadian Basin. The net impact is complementary to the nutrient impact carried into the BG interior by deeper, cold-core eddies[69].

**Outlook.** The novel observations presented here reveal a jet of warm salty water that appears to be (1) subducting beneath a cool fresh surface layer, and (2) vertically compressing and breaking

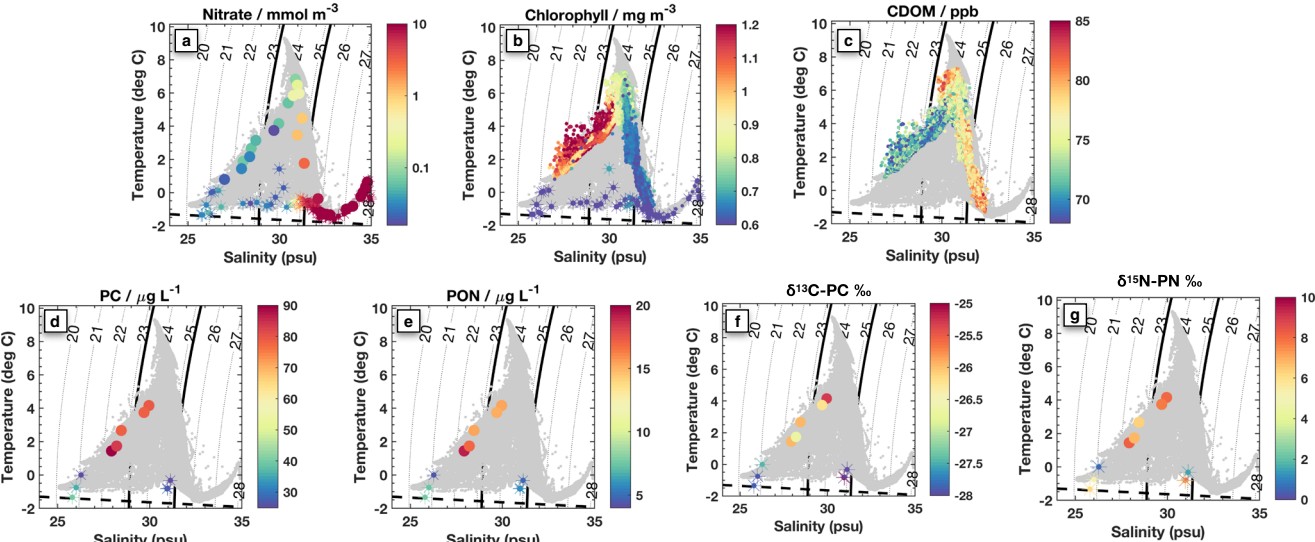

**Fig. 7 Biogeochemical signatures of Pacific Summer Water (PSW): temperature–salinity (T-S) patterns.** T-S panels in the same style as Fig. 3 are colored by **a** Nitrate concentration, **b** Chlorophyll, **c** CDOM, **d** particulate carbon (PC) concentration, **e** particulate organic nitrogen (PON) concentration, **f** particulate carbon isotopes, **g** particulate organic nitrogen isotopes. In **b**, **c** small dots are from the Wirewalker measurements, and in all panels larger symbols are from ship-board CTD bottle samples taken from casts that were conducted at locations where the warm jet was (filled circles) and was not (stars) present.

up into smaller, spinning eddies. Though some heat is lost to the atmosphere, the vast majority subducts to be sequestered from direct contact with the atmosphere, and subsequently stirred and spread sub-surface toward the central basin. Both the subduction and development of relative vorticity are qualitatively consistent with cross-jet circulations associated with mesoscale jet meanders, submesoscale instabilities, and potential vorticity conservation.

Meandering warm jets offshore of Barrow Canyon are not unusual[33,39,40,60]. In previous studies, some of that heat leaving the coast near Barrow Canyon seems to be firmly ejected offshore, while some meanders to rejoin one of the boundary currents, and may subsequently leak into the interior by other stirring processes[62]. The subduction and intrathermocline eddy development processes observed here may be a regular occurrence, linking the larger-scale influxes of warm, surface-intensified PSW observed within Barrow Canyon[21,29] with discrete intrathermocline eddies and filaments observed within the basin[17,18].

The net effect on redistributing heat and biogeochemical properties in the main basin will depend on the interplay between lateral stirring, vertical mixing, and the evolving large-scale basin circulation. How long warm intrathermocline eddies last, and thus how far they get under the sea ice, is an open question involving their translation speed, intrinsic stability[59], and frictional effects of the ice above[36,70]. While lateral transport of shelf-origin waters helps maintain the upper Arctic halocline and its nutrient maxima[71], upward nutrient fluxes through the halocline depend both on halocline ventilation rates and lateral fluxes within the surface layer; differential changes may alter both biological production and biogeochemical cycling rates[72].

As the heat content of PSW is growing[8,14,21], the combination of PSW subduction, lateral stirring, and upward vertical mixing should lead to a pattern of accelerating sea ice melt spreading out from the Pacific inflow, as has been observed in recent decades[6,73]. Related processes have been observed on the other side of the Arctic, associated with warming of subducted Atlantic water[74]. At the same time, the BG is spinning up, with a growing accumulation of near-surface freshwater that tends to deepen the halocline and nitricline[75,76]. However, as the PSW warms and lightens, it is subducting along shallower isopycnals, putting it

closer to the surface[77]. The physical insights gained with this work should help develop more accurate ways to represent the subduction and stirring processes in models. Improved understanding and modeling ability of the processes described here will help forecast the detailed geography and time-frame for the changing ecosystem and accelerating Arctic sea ice loss.

## Methods

Most of the measurements presented here were conducted during September 2018 aboard the R/V *Sikuliaq*, operated by the University of Alaska. Details of each measurement technique, as well as the analysis calculations presented above, are shown here. Additional views of the intermediates steps of these calculations are presented in Supplementary material.

**Satellite image.** The hybrid MODIS image in Fig. 1b was created by combining a Level 2 SST product created by the Ocean Biology Processing Group, NASA Goddard Space Flight Center, and a true-color image created at the University of Miami by applying the NASA Blue Marble algorithm to MODIS bands 1, 4, and 3. The two products were interpolated to a common spatial grid, and a threshold was applied to the red band of the interpolated true-color image. Pixels above this threshold, corresponding to sea ice, land, and clouds, were assigned their true color in the hybrid image. Color for the below-threshold pixels, corresponding to open ocean surface, was assigned using the temperature scale shared with the sub-surface measurements in Fig. 1c.

**Fast CTD.** The Fast CTD (conductivity-temperature-depth) system is a rapid profiler built and operated by the Multiscale Ocean Dynamics Group at the Scripps Institution of Oceanography, UC San Diego. The system was operated from a custom direct drive electric winch mounted on the starboard quarter of the ship, and profiled off a 10 m long boom that is deployed outboard and aft, in order to avoid contamination by the ship wake. With the very low drag FCTD fish, vertical profiling speeds up to 5 m/s were conducted while the ship was steaming at up to 5 knots. Temperature, conductivity, and pressure measurements were made with a Seabird SBE49 instrument, sampled at 16 Hz, and binned to 0.5 m vertical resolution. Temperature and conductivity data were adjusted to match phase and resolution before computing salinity, using standard techniques. Profiles to 200 m depth were completed every few minutes, leading to an average horizontal spacing of 160 m between profiles. Salinity and water density were calculated from temperature and conductivity using standard routines.

**Wirewalker.** A Wirewalker drifting, wave-powered profiling system[78] conducted one deployment during this survey (Fig. 1c, red track). The vehicle is equipped with CTD (RBR Concerto), velocity (Nortek Signature 1 MHz), optics (WETLabs chlorophyll and CDOM fluorescence, 532 nm backscatter), and a microstructure sensor. The

surface buoy is outfitted with a satellite tracker. The profiler was deployed on a 100-m wire. Profiling speed depended on the wave state, with the average profiling frequency 7.6 cycles per hour. The pathway of Wirewalker drift is shown in Fig. 1c. The drift lasted from 07:30 on 15 Sept 2018 to 06:30 on 16 Sept 2018. Wirewalker chlorophyll data have been corrected for non-photochemical quenching[79]. Although water bottle samples for CDOM were not collected, the vertical gradients shown in Fig. 6 are an order of magnitude larger than the instrument sensitivity (~0.2 ppb) and therefore reflect actual environmental variability.

**Ocean velocity.** Ocean currents were measured with a 300 kHz Acoustic Doppler Current Profiler from Teledyne RDI, installed in the centerboard drop keel of the *Sikuliaq*. Data were sampled with 1 Hz and 2 m vertical resolution. Data presented in Figs. 1 and 2 has been smoothed and sub-sampled to 2 min temporal resolution. Velocity at this resolution has an uncertainty of ±2 cm/s due to instruemnt noise. The shear data shown in Fig. 2 have been further smoothed to 4 min and 4 m resolution, producing an estimated shear uncertainty of $0.003\,s^{-1}$.

**Barrow Canyon data.** Temperature and salinity profile data from September 2018 used in Fig. 3a come from a hydrographic database that was compiled and quality-controlled as described by[8] with additional data from an array of autonomous ocean profilers deployed jointly by the Woods Hole Oceanographic Institution and the NOAA Arctic Heat experiment[80].

**Ship-board fluxes.** Turbulent fluxes over open water were computed from ship meteorological sensors using the COARE 3.5 algorithm[81] as refined by[82], using the cool-skin option. Additional details are in the Supplementary material.

**Turbulent mixing.** Upward turbulent heat fluxes are calculated from the Modular Microstructure Profiler (MMP, track shown in Fig. 1c, blue line). MMP is a loosely tethered free-fall instrument ballasted to at $0.7\,m\,s^{-1}$. As the instrument falls, air-foil shear probes measure shear with centimeter resolution[83,84]. The MMP was profiled from the FastCTD winch/boom system, repeatedly cycled to a depth of 100 m. Turbulent dissipation rates ($\epsilon$) were computed from the shear data using a frozen field hypothesis and standard spectral fitting techniques[85], with a noise level near $10^{-10}$ W $kg^{-1}$. The MMP is also equipped with a pumped Seabird CTD for measuring temperature, conductivity, and pressure, from which salinity and potential density are calculated using standard techniques. Turbulent heat flux ($J_q$) was calculated as $J_q = -\rho c_p \frac{0.2\epsilon}{N^2}\frac{dT}{dz}$, where $\rho$ is the ocean density, $c_p$ the specific heat of seawater, $N^2$ the buoyancy frequency, and $T(z)$ the measured vertical profile of temperature. See Supplementary for further details.

**SWIFT drifters.** SWIFT drifters provide estimates of the dissipation rate of turbulent kinetic energy in the upper few meters of the surface ocean[86]. Profiles of turbulent velocity fluctuations are measured using the center beam of a down-looking Nortek Signature1000 in the hull of the SWIFT, using a pulse-coherent "HR" mode with 4 cm bin size and 8 Hz sampling. These profiles are used to compute a spatial structure-function, from which the dissipation rate is estimated by fitting an $r^{2/3}$ dependence ($r$ is the distance between bins). These estimates are produced every 12 min, in a surface wave-following reference frame. Conductivity and temperature are measured in situ at the upper most bin (0.3 m below the surface).

**Particulate carbon and nitrogen isotopes.** For isotopic analysis, seawater was collected using Niskin bottles attached to a CTD rosette frame. Between 1.8 and 3.7 L of seawater was filtered onto pre-combusted glass fiber filters (nominal pore size of 0.7 micron) using vacuum filtration. Filters were stored frozen, then freeze-dried for 12 h and pelletized. Filters were analyzed at the University of Liverpool using a Costech Elemental Analyzer coupled to a Thermo isotope ratio mass spectrometer. Particulate carbon and particulate organic nitrogen concentrations and stable carbon and nitrogen isotope values were corrected using international reference standards (USGS40 and USGS41A) and expressed in $\delta$ notation ($\delta^{13}$C-PC (‰ vs VPDB) = $(R_{sam}/R_{std} - 1) \times 1000$ and $\delta^{15}$N-PN (‰ vs AIR) = $(R_{sam}/R_{std} - 1) \times 1000)$. Standards were run in triplicate with a reproducibility better than 0.1‰ for carbon and 0.05 ‰ for nitrogen.

**Nutrients.** Samples for the analysis of dissolved inorganic nutrients (nitrate, nitrite, ammonium, phosphate and silicate) were collected from multiple depths resolving full-depth profiles. The location of the casts is indicated in white dots in Fig. 3g, and presented in Table 1 of the Supplementary material. 100 mL of seawater were collected directly from Niskin bottles into 120 mL HDPE bottles (10% HCl pre-cleaned) using an inline AcroPak (0.45 μm pore size) filter. Samples were then stored frozen (upright) at −20 °C for later analysis on land. Analyses were carried out at the Marine Chemistry Laboratory, University of Washington, using an AA3 Seal Analytical continuous seg-mented flow nutrient analyzer following standard colorimetric techniques. Overall precision and accuracy were equal or better than 2%, as assessed via measurements of OSIL Scientific certified materials. For the purpose of the current study, only nitrate data are shown.

**Section calculations.** Lateral heat flux carried by the warm jet was calculated as $\rho c_p U(T - T_f)$ where $\rho$ is the ocean density, $c_p$ the specific heat, $U$ the along-jet velocity, and $T_f$ is the salinity dependent freezing point of seawater[8,14,21].

In Fig. 2, the vertical shear ($dU/dz$) is calculated by first-differencing along-jet velocity ($U$), and then smoothing with box-car filters of 425 m horizontally and 4 m vertically. Here the shear has been scaled by the quantity $f_c = f - 2*(U/r)$, i.e., the local inertial frequency $f$ adjusted by twice the curvature vorticity of the jet. This term accounts for the form of thermal wind balance expected for cyclogeostrophic flows[57–59]. We use the best fit for curvature at this point, though the sign and magnitude of the curvature varies throughout the meandering process.

The lateral buoyancy gradient is calculated by first-differencing buoyancy ($b = -(g/\rho_0)\rho'$ where $\rho_0 = 1025$ kg m$^{-3}$ is a reference density, and $\rho'$ is the potential density calculated from temperature and salinity) in the across-jet direction ($y$), smoothing with the same box-car filters. Lateral gradients in along-jet velocity ($dU/dy$) are calculated similarly to those of buoyancy. The Rossby number ($Ro = [-dU/dy - U/r]/f$) takes into account the curvature of the meandering jet. To estimate $r$ we use the best fit for the radius of curvature of the flow, $r = 30$ km, at this point.

The Ertel Potential Vorticity (PV, Fig. 4) is a conserved quantity of an inviscid, adiabatic fluid in a rotating reference frame[42]. Following previous work for currents with a appreciable curvature[57,58] we write the three terms comprising the PV as $PV = f\frac{\partial b}{\partial z} - \left(\frac{\partial U}{\partial y} + \frac{U}{r}\right)\frac{\partial b}{\partial z} + \frac{\partial U}{\partial z}\frac{\partial b}{\partial y}$, which from left to right are the quantities labeled squashing, spinning, and tilting in Fig. 4a–c.

The ageostrophic cross-jet velocity in Fig. 2g is calculated by numerically solving the adiabatic, two-dimensional version of the quasigeostrophic omega equation $f^2\frac{\partial^2 \psi}{\partial z^2} + \frac{\partial b}{\partial z}\frac{\partial^2 \psi}{\partial y^2} = 2\frac{\partial V}{\partial y}\frac{\partial b}{\partial y}$ where $\Psi$ is the streamfunction, $V$ the cross-jet velocity. The omega equation was solved over the domain shown in Fig. 2 using boundary conditions of $\frac{\partial \psi}{\partial x} = 0$ at the top and bottom boundaries and $\frac{\partial \psi}{\partial z} = 0$ at the side boundaries, using the Matlab minimal residual method solver with a tolerance of $10^{-4}$, following[53]. From the streamfunction, ageostrophic cross-front and vertical velocities are calculated as $v_a = \frac{\partial \psi}{\partial z}$, $w_a = -\frac{\partial \psi}{\partial y}$ respectively. Cross-jet velocity confluence ($dV/dy$) was extrapolated down to 130 m by assuming it had the same vertical decay as lateral buoyancy gradients, both of which characterize the mesoscale flow. The lower boundary of 130 m is somewhat arbitrary. Imposing a lower boundary at the base of the measured velocity range (~90 m) creates an artificial and unrealistic lower boundary to the streamfunction; density measurements show the sloping front extends deeper. However extrapolating velocity data below the range of measurements is also artificial. Sensitivity tests were performed using lower boundary conditions of 100–200 m, in 20 m increments. For deeper lower boundaries, the magnitude of the cross-jet circulation increases, but all simulations show the same qualitative pattern as in Fig. 4. Given that the sensitivity in magnitude of cross-front circulation strength is likely smaller than the intrinsic caveats associated with using this method for a Ro ~ 1 flow, we emphasize the qualitative pattern.

## Data availability

• Arctic Heat data are available through https://www.pmel.noaa.gov/arctic-heat/data

• All ship-board and mooring data collected as part of the US Office of Naval Research Stratified Ocean Dynamics of the Arctic program is still in the process of being organized and archived. All data will be fully released to the public when the program formally concludes. In the meantime, details about the program and data collected can be seen here (http://www.apl.washington.edu/project/project.php?id=soda). SODA data presented in this study are available from the corresponding author upon reasonable request.

• Data from the ARISE, PEANUTS, and BMBF projects are also available from the corresponding author upon reasonable request.

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

## Acknowledgements
Support for this work was provided by the US Office of Naval Research Stratified Ocean Dynamics of the Arctic program (grant numbers N000141512903, N000141612378, N000141612377, N000141612379, N0001416123450, N000141612360, N000141612349, N000141812007, N000141812475, and N000141912514). Additional support for bio-geochemistry sampling was provided by UK (NERC) and Germany (BMBF) through the Changing Arctic Ocean Program's ARISE (NE/P006035/1, NE/P006000/2), PEANUTS (NE/R01275X/1, NE/R012547/2, and BMBF 03F0804) projects and the UK-France PhD program DGA/Dstl. Float deployments and hydrographic data compilations were supported in part by North Pacific Research Board grants A91-99a and A91-00a. The deployment of autonomous ocean profilers was supported by ONR, NOAA Research, and the Joint Institute for the Study of the Atmosphere and Ocean (JISAO) under NOAA Cooperative Agreement NA15OAR4320063. We are grateful to the engineers within our research groups and the captain and crew of the R/V *Sikuliaq* for facilitating this work, particularly Mike Goldin, Ethan Roth, Cris Seaton, and Paul St.Onge.

## Author contributions
Experimental design for this survey was led by J.A.M., H.L.S., J.H., J.T., and T.P. Instrument design and development for the primary survey was led by M.H.A., A.J.L., and J.T. Data acquisition on the primary cruise was carried out by J.A.M., H.L.S., J.H., J.T., T.P., B.I.B., S.B., N.C., S.D., E.C.F., J.G., T.K., C.J., and M.M.S. Satellite data acquisition and analysis were carried out by J.H., H.C.G., and B.L. Mooring development, deployment, and analysis were carried out by C.L., L.R., and S.D.B. Glider and drifter deployments and analysis were led by S.L.D., S.R.J., and K.R.W. Biogeochemistry bottle samples were collected, processed and analyzed by B.I.B., Y.D.L., J.E.H., C.M., L.N., and S.T.V. Instability calculations were carried out by J.A.M. and L.N.T. C.L. is the lead PI for the SODA project. All authors contributed to writing and editing the manuscript.

## Competing interests
The authors declare no competing interests.
