## [Peer Review File · Nature Communications]

REVIEWERS' COMMENTS

Reviewer #4 (Remarks to the Author):

I have read "A warm jet in a cold ocean: subduction and heat storage in the Western Arctic" by MacKinnon et al. in consideration for a publication in Nature Communications. The paper presents a detail oceanographic survey on the inflow of the warm Pacific-origin water into the Arctic Ocean. The Arctic Ocean receives inflows from the Atlantic and Pacific Oceans, and it is important to understand how the incoming heat from these two oceans influence the state of the Arctic Ocean. The inflow in the Pacific sector is relatively not well understood compared to the inflow in the Atlantic side. While this paper presents a wealth of interesting observations, I have some hesitations in the context of the presentation. The paper describes the subduction mechanism based on the existing knowledge, which makes the readers think that this type of the detail survey has been done elsewhere. I believe that the importance for the readers of Nature Communications is that if this mechanism has any implications to the state of the Arctic Ocean (e.g. sea ice, primary production), which this paper does not make a clear statement. I would have appreciated if the authors described the variability of the jet with respect to variables that have reliable long-term measurements, such as wind, and speculated the role of the jet in the context of the last 40 years. In the present form, the novelty in this paper is limited in describing the subduction process from a snapshot near Barrow Canyon and showing that the existing theoretical framework can be used to assess the structure of the jet offshore. As a result, the audience is targeted to oceanographers who are interested in the Western Arctic. I believe that this publication shall be considered in other more creditable and discipline specific journals, such as Journal of Geophysical Research.

Reviewer #5 (Remarks to the Author):

Thank you for the opportunity to review this paper. I have been asked by the editor to join the other reviewers at this later stage, and to comment on the extent to which the authors have addressed the previous comments.

I personally find the observations presented in this work of interest. The mechanisms these observations show -- namely, the mechanisms driving heat and biogeochemical tracers fluxes across the shelf of the Canadian Basin -- are of importance for the evolution of the larger Arctic Ocean and its ice cover.

I share the previous reviewers' opinion that it is (remains) challenging to understand the full story. In my opinion there is too much emphasis on the theory behind physical processes driving subduction, and on speculations about their effects. As the authors discuss themselves, the theory has already been addressed by other studies. I also share reviewer's 1 opinion regarding the difficulties to discuss general mechanisms based on observations of a single event. In the limited space allowed, this is to the detriment of a clearer presentation of the observations, which are the main strength of this paper. I would suggest the authors to focus more on this strength, and in particular on where in the various figures can we see the different processes discussed. These are informative but complex figures showing many different processes; more guidance on where to find them would make the paper much more accessible.

Some specific suggestions that, in my opinion, could make the manuscript clearer:

- I find the section "subduction processes" particularly challenging. A reader not versed in the field would find quite difficult to follow the discussion. For example, the discussion of the omega equation, its validity, and its limitations contains a lot of jargon; the omega equation itself does not appear in the main text, despite being referenced for an entire page. I understand this section has been added in response to the previous reviews, but I wonder if moving most of it to the method section could help streamlining the discussion? A schematic, or a more intuitive description of the different

processes considered and where these can be seen in the figures, would help the reader navigating the many concepts introduced. I would find useful to add a succinct list of all processes considered somewhere in the text.

- Line 100: "range of hypothesis for the types of instabilities" this could be a good place to list the processes that will be considered, and later discussed in the figures.

- Paragraph starting on line 105: I think the paper would be stronger if focused on showing observational evidences of the detailed physical processes, rather than trying improve our understanding of them -- the latter has been done in other studies.

- Line 131: "Within the PSW" I would point again the reader to where the PSW is in the figures.

- Line 136: "A cross sectional slice" I would point to its location on Figure 1c

- Line 138: contours are white, not black

- Paragraphs starting on line 151 and 158: I think a reader from outside the field would find these paragraphs difficult

- Figure 2: adding South and North arrows to the panels, as well as an arrow indication the direction of the Ekman transport, would be helpful.

- Line 187: "shown in Figure 4e,f" by contours or by colors? Again, a reader from outside the field might find more guidance helpful

- Line 192: "(arrows in Fig. 4e)" I would add "cyan", and "pink arrows" at the end of the sentence.

- Line 206: "Ro is close to -1" I would add "(see Figure 2g)".

- Figure 4: the order of the lettering in the panels is off (a c b d e f)

- Line 241: it repeats what has just been said at the end of the previous paragraph

- Line 321: a dynamical explanation for the maintenance of subsurface eddies below ice has been provided by Ou and Gordon (1986) <http://dx.doi.org/10.1029/JC091iC06p07623> .

- Line 379 and line 171: should the authors be interested, I will take the liberty to suggest a recent paper in which we discuss a dynamical explanation for the growth and persistence of subsurface eddy fluxes in the seasonally ice covered Arctic <https://doi.org/10.1175/JPO-D-20-0054.1>

Summarizing, I find the paper interesting, but I would suggest it would be worth trying to make it even more accessible by focusing more on connecting the processes discussed with where they can be seen in the figures and less on theoretical discussion which have already been reported elsewhere.

I do hope the authors will find my suggestions useful. They are of course only suggestions, and a more accessible manuscript can be achieved in other ways.

Thank you again.

Best

Gianluca Meneghello

Revision Responses

Reviewer #4

I have read "A warm jet in a cold ocean: subduction and heat storage in the Western Arctic" by MacKinnon et al. in consideration for a publication in Nature Communications. The paper presents a detail oceanographic survey on the inflow of the warm Pacific-origin water into the Arctic Ocean. The Arctic Ocean receives inflows from the Atlantic and Pacific Oceans, and it is important to understand how the incoming heat from these two oceans influence the state of the Arctic Ocean. The inflow in the Pacific sector is relatively not well understood compared to the inflow in the Atlantic side. While this paper presents a wealth of interesting observations, I have some hesitations in the context of the presentation. The paper describes the subduction mechanism based on the existing knowledge, which makes the readers think that this type of the detail survey has been done elsewhere. I believe that the importance for the readers of Nature Communications is that if this mechanism has any implications to the state of the Arctic Ocean (e.g. sea ice, primary production), which this paper does not make a clear statement.

I would have appreciated if the authors described the variability of the jet with respect to variables that have reliable long-term measurements, such as wind, and speculated the role of the jet in the context of the last 40 years. In the present form, the novelty in this paper is limited in describing the subduction process from a snapshot near Barrow Canyon and showing that the existing theoretical framework can be used to assess the structure of the jet offshore. As a result, the audience is targeted to oceanographers who are interested in the Western Arctic. I believe that this publication shall be considered in other more creditable and discipline specific journals, such as Journal of Geophysical Research.

Thanks for all these comments. All science starts from some basis of existing knowledge, but it is often a confusing array of possibility. In a general sense there are lots of different mechanisms that allow or facilitate subduction of one water mass under another in the ocean. Some theories appear to accurately describe observations in some situations or places in the world, but not others. There have also been a number of theoretical predictions and hypotheses as to how the process works in the Arctic Ocean specifically, but we have been lacking clear observational evidence to differentiate one hypothesis from another. We feel the observations presented here thus represent a significant step forward in allowing us to pin down some of the involved mechanisms. That new understanding the paves the way for improved representation of the process in forecast models. Given that these sub-surface pockets of heat are playing an increasingly important role in accelerating Arctic sea ice melt, we feel our results are of broad interest, impact and significance, and appropriate for Nature Communications.

We also appreciate the suggestion to consider the variability of these incoming warm water jets over multiple years. We cite and describe several previous papers showing jets like this are not uncommon in the summer - this is not a one-off event. A thorough description of how such features evolve from year to year would also be an excellent contribution, but beyond the scope of this paper.

Reviewer #5

Thank you for the opportunity to review this paper. I have been asked by the editor to join the other reviewers at this later stage, and to comment on the extent to which the authors have addressed the previous comments.

I personally find the observations presented in this work of interest. The mechanisms these observations show -- namely, the mechanisms driving heat and biogeochemical tracers fluxes across the shelf of the Canadian Basin -- are of importance for the evolution of the larger Arctic Ocean and its ice cover.

I share the previous reviewers' opinion that it is (remains) challenging to understand the full story. In my opinion there is too much emphasis on the theory behind physical processes driving subduction, and on speculations about their

effects. As the authors discuss themselves, the theory has already been addressed by other studies. I also share reviewer's 1 opinion regarding the difficulties to discuss general mechanisms based on observations of a single event. In the limited space allowed, this is to the detriment of a clearer presentation of the observations, which are the main strength of this paper. I would suggest the authors to focus more on this strength, and in particular on where in the various figures can we see the different processes discussed. These are informative but complex figures showing many different processes; more guidance on where to find them would make the paper much more accessible.

We thank the reviewer for this suggestion. We have gone through the paper and added significant additional detail and description specifically commenting on how the processes discussed show up in the various figure. We hope that these changes will indeed make the paper more accessible.

Some specific suggestions that, in my opinion, could make the manuscript clearer:

- I find the section "subduction processes" particularly challenging. A reader not versed in the field would find quite difficult to follow the discussion. For example, the discussion of the omega equation, its validity, and its limitations contains a lot of jargon; the omega equation itself does not appear in the main text, despite being referenced for an entire page. I understand this section has been added in response to the previous reviews, but I wonder if moving most of it to the method section could help streamlining the discussion? A schematic, or a more intuitive description of the different processes considered and where these can be seen in the figures, would help the reader navigating the many concepts introduced.

This is a great point, and a persistent challenge in presenting material like this that has both a broad big-picture impact, and some intrinsic complexity in the details. After pondering a while in light of your suggestions, we have moved some of the more technical parts of this discussion, including most details of Centrifugal instability and turbulent thermal wind, to the Supplementary material. Readers who are interested in those details can find them there, and the paper flows a bit better now (hopefully!). We have also made an effort to go through and add more detail highlighting how these types of processes show up specifically in the observations, which we agree is a strength of this work.

I would find useful to add a succinct list of all processes considered somewhere in the text. Line 100: "range of hypothesis for the types of instabilities" this could be a good place to list the processes that will be considered, and later discussed in the figures.

We have rewritten both the introductory and closing text to more clearly present and recapitulate the relevant processes.

- Paragraph starting on line 105: I think the paper would be stronger if focused on showing observational evidences of the detailed physical processes, rather than trying improve our understanding of them -- the latter has been done in other studies.

Validation of which of various processes proposed by theory and numerical studies are actually observed in the real ocean is an important part of that understanding - the text has been amended to clarify that this is what we mean here.

- Line 131: "Within the PSW" I would point again the reader to where the PSW is in the figures.

Thanks, done

- Line 136: "A cross sectional slice" I would point to its location on Figure 1c

Good idea, done

- Line 138: contours are white, not black

Fixed

- Paragraphs starting on line 151 and 158: I think a reader from outside the field would find these paragraphs difficult

Agree, these have been rewritten in slightly plainer language to make the content more accessible.

- Figure 2: adding South and North arrows to the panels, as well as an arrow indication the direction of the Ekman transport, would be helpful.

We have added north and south arrows to Figure 2, thanks for that suggestion. The Ekman discussion has been primarily moved to the Supplementary material.

- Line 187: "shown in Figure 4e,f" by contours or by colors? Again, a reader from outside the field might find more guidance helpful

Thanks, we have rewritten to be more accessible

- Line 192: "(arrows in Fig. 4e)" I would add "cyan", and "pink arrows" at the end of the sentence.

Done

- Line 206: "Ro is close to -1" I would add "(see Figure 2g)".

Done, although some of this content has been moved to Supplementary

- Figure 4: the order of the lettering in the panels is off (a c b d e f)

Fixed

- Line 241: it repeats what has just been said at the end of the previous paragraph

Good point, fixed

- Line 321: a dynamical explanation for the maintenance of subsurface eddies below ice has been provided by Ou and Gordon (1986) <http://dx.doi.org/10.1029/JC091iC06p07623> .

Thanks, we have added that reference

- Line 379 and line 171: should the authors be interested, I will take the liberty to suggest a recent paper in which we discuss a dynamical explanation for the growth and persistence of subsurface eddy fluxes in the seasonally ice covered Arctic <https://doi.org/10.1175/JPO-D-20-0054.1>

Thanks, enjoyed reading this paper, we have added mention of it to our conclusions.

Summarizing, I find the paper interesting, but I would suggest it would be worth trying to make it even more accessible by focusing more on connecting the processes discussed with where they can be seen in the figures and less on theoretical discussion which have already been reported elsewhere.

We appreciate the focus on accessibility. As mentioned above, significant effort has been undertaken to the main text more consistent and less jargon-y. We feel this has definitely improved the paper, so appreciate the impetus to do so.

I do hope the authors will find my suggestions useful. They are of course only suggestions, and a more accessible manuscript can be achieved in other ways.

We do find them useful thank you!